# Investigation of Excited-State Intramolecular Proton Transfer and Structural Dynamics in Bis-Benzimidazole Derivative (BBM)

**DOI:** 10.3390/ijms24119438

**Published:** 2023-05-29

**Authors:** Junhan Xie, Ziyu Wang, Ruixue Zhu, Jiaming Jiang, Tsu-Chien Weng, Yi Ren, Shuhua Han, Yifan Huang, Weimin Liu

**Affiliations:** 1School of Physical Science and Technology, ShanghaiTech University, Shanghai 201210, China; 2Key Lab of Colloid and Interface Chemistry, Ministry of Education, Shandong University, Jinan 250100, China; 3STU and SIOM Joint Laboratory for Superintense Lasers and the Applications, Shanghai 201210, China

**Keywords:** excited-state intramolecular proton transfer, structural dynamics, transient absorption, femtosecond fluorescence upconversion, femtosecond stimulated Raman spectroscopy

## Abstract

The bis-benzimidazole derivative (BBM) molecule, consisting of two 2-(2′-hydroxyphenyl) benzimidazole (HBI) halves, has been synthesized and successfully utilized as a ratiometric fluorescence sensor for the sensitive detection of Cu^2+^ based on enol–keto excited-state intramolecular proton transfer (ESIPT). In this study, we strategically implement femtosecond stimulated Raman spectroscopy and several time-resolved electronic spectroscopies, aided by quantum chemical calculations to investigate the detailed primary photodynamics of the BBM molecule. The results demonstrate that the ESIPT from BBM-enol* to BBM-keto* was observed in only one of the HBI halves with a time constant of 300 fs; after that, the rotation of the dihedral angle between the two HBI halves generated a planarized BBM-keto* isomer in 3 ps, leading to a dynamic redshift of BBM-keto* emission.

## 1. Introduction

Derivatives of 2-(2′-hydroxyphenyl) benzimidazole (HBI) and its analogs 2-(2′- hydroxyphenyl) benzoxazole (HBO) and 2-(2′-hydroxyphenyl) benzothiazole (HBT) have been widely used for metal ions Zn^2+^ and Cu^2+^ selective emission ratiometric probe due to large Stokes shift and dual fluorescence emission based on enol → keto excited-state intramolecular proton transfer (ESIPT) [1,2,3,4,5]. When the molecules sufficiently bind to the metal cations, these cations act as ligands and can disrupt the ESIPT process, simultaneously resulting in a decrease in keto emission and an increase in enol fluorescence emission [1]. It has been demonstrated that the ESIPT processes of hydrogen bonding molecules having a hydroxyl group as a proton donor, such as HBT [6,7,8,9], HBO [10,11], and HBI [12,13], are usually highly exergonic with a low barrier or barrierless, leading to remarkably ultrafast proton transfer. Despite such a unique application, the efficiency of the large Stokes shift keto emission is generally low due to cis to trans photoisomerization via the C=C bond between the benzene ring and five-membered heterocycles [9], which significantly affects the detection effect of the proportional fluorescent probe. An in-depth understanding of the photophysics of these kinds of molecules is critical to determining the underlying mechanisms for successful metal ion probes in cell biology. The behaviors of ESIPT and photoisomerization have been investigated by time-resolved spectroscopies, including fluorescence upconversion measurements and broadband femtosecond transient absorption spectroscopy [9,14,15,16,17]. These photophysical studies indicate that solvent-dependent ESIPT rates vary from a hundred femtoseconds to a few picoseconds [9,14,18]. However, the details of the photoisomerization of those molecules are still not well-understood. By using the ultrafast Raman loss spectroscopy (URLS) technique, Umapathy et al. demonstrated direct evidence for the involvement of torsional motion leading to an ultrashort lifetime of HBT, which indicates the evolution of the planar cis-keto* form to the twisted keto* form [19].

In this study, we combined tunable femtosecond stimulated Raman spectroscopy (FSRS) [20,21,22,23,24,25], transient absorption (TA), and fs fluorescence upconversion spectroscopy to comprehensively elucidate the ESIPT and conformational change of the bis-benzimidazole derivative (BBM) molecule, which consists two HBI derivatives (see Figure 1) and has been successfully used as a ratiometric fluorescence sensor for the sensitive detection of Cu^2+^ [26]. Different from the HBI molecule, BBM exhibits unique ESIPT and intramolecular twisting dynamics.

## 2. Results and Discussion

### 2.1. Steady-State Absorption and Fluorescence Spectroscopy

The normalized absorption and emission spectra of BBM in tetrahydrofuran (THF; relative polarity: 0.2; viscosity: 0.55 cP) and dimethyl sulfoxide (DMSO; relative polarity: 0.44; viscosity: 2.24 cP) are presented in Figure 1a. The absorption band of BBM is observed at ~350 nm in both solvents and is ascribed to the π–π* transition between benzimidazole and hydroxyphenyl. Two maxima at 340 and 350 nm are assigned to the anti-enol and syn-enol BBM conformers, respectively [27,28]. Notably, the peaks corresponding to the anti-enol in THF exhibit less distinctness than those observed in DMSO. This phenomenon can be attributed to a solvent-dependent conformational equilibrium between syn- and anti-enols, which results in varying proportions of these species [28]. The BBM fluorescence spectra exhibit a significant Stokes shift of 86 nm in THF and 95 nm in DMSO versus the absorption peak, indicating keto emission after ESIPT. Furthermore, a weak shoulder at 390 nm was observed, which is attributed to fluorescence emission of the BBM-enol* form (see Appendix A). The bathochromic spectral shift of the keto emission with respect to the polar solvents can be interpreted as the effect of solvent polarity on the excited electron state BBM-keto*.

### 2.2. Femtosecond Transient Absorption Spectroscopy

Femtosecond transient absorption (TA) measurements were first performed to obtain the excited-state dynamics of the BBM molecule. Figure 2a and Appendix A show the ultrafast two-dimensional TA spectra of BBM molecules in DMSO and THF under excitation at 350 nm with a pulse energy of 70 nJ/pulse. The TA spectrum against the delay time from −1 ps to 8 ns exhibits intricate excited-state features between 350 nm and 650 nm including three excited-state absorption bands (ESA1 band center at 380 nm, ESA2 band center at 440 nm, and ESA3 band of 460 nm–640 nm), the stimulated emission band (SE center at 450 nm), and the ground-state bleach at the blue edge. Moreover, global analysis with a parallel model was employed to obtain the decay-associated difference spectra (DADS; see Figure 2b), yielding four characteristic lifetimes with time constants of τ_1_ = 0.3 ± 0.2 ps, τ_2_= 3.0 ± 0.4 ps, τ_3_ = 1.6 ± 0.2 ns, and τ_4_ = infinitely long lifetime in DMSO.

As shown in Appendix A, between the 0 and 0.6 ps timescales, the BBM temporal trace in DMSO at 425 nm (ESA2 band) exhibits initial rapid decay dynamics with a time constant of 0.3 ps, and the ESA3 band at 540 nm significantly rises on the same timescale (see Figure 2c). This evolution is attributed to the ESIPT from the excited state of BBM-enol* to BBM-keto*. After 0.6 ps, as shown in Figure 2d, the ESA3 band at 540 nm exhibits a dominant two-exponential decay with time constants of τ_2_ = 3.0 ps and τ_4_ = infinitely long lifetime. Notably, the decay of the ESA3 band is followed by the increase in the ESA1 band signal at 383 nm in 3.0 ps (see Figure 2d). The ESA1 band at 383 nm then decays with two exponential processes of τ_3_ = 1.6 ns and τ_4_ = infinitely long lifetime (see Appendix A). The SE band at 450 nm reveals 1.6 ns decay dynamics indicating that the 1.6 ns is attributed to the spontaneous emission lifetime from the excited state of BBM-keto*, which was further confirmed using the fs upconversion fluorescence spectroscopy at an excitation of 350 nm, as shown in Figure 3. We utilized EOS-TA measurements in the ns ~ μs time range to determine the longest-lived components τ_4_ in TA spectra. As shown in Appendix A, a broad ESA band in the range of 370–850 nm exhibits single exponential decay with a time constant of 66 ± 2 ns, indicating depopulation from BBM-keto* triplet T_1_ state [9,29].

Previous reports have demonstrated that the significant role of the twisting motion pivot on the C23–C25 (C14–C15) bond was observed from HBI-keto*, which results in fluorescence quenching via cis–trans photoisomerization [6,9,30]. To probe possible photoisomerization of BBM in the course of the primary photo-processes, the sample was measured in THF with low viscosity using TA. As shown in Appendix A, the DADS of BBM in THF reveal four exponential decay components of 0.3 ± 0.1 ps (τ_1_), 2.6 ± 0.3 ps (τ_2_), 1.7 ± 0.3 ns (τ_3_), and an infinite lifetime (τ_4_); the longest τ_4_ time components are well-resolved with a time constant of 862 ± 140 ns, which is assigned to the triplet keto-T_1_ state lifetime (see Appendix A). The spectral shape of each DADS component does not differ from that in the DMSO solvent, indicating that the intermediate states of the BBM molecule remain unchanged in THF (see Figure 2b and Appendix A). Notably, the second component τ_2_ has shorter lifetimes in the low viscous THF solvent versus that in DMSO, indicating that a conformational change occurs after ESIPT processes.

### 2.3. Femtosecond Fluorescence Upconversion Spectroscopy

Our TA spectra, thus far, suggest enol to keto ESIPT, which is followed by the photoinduced conformational change of the BBM chromophore. However, two open questions here remain unclear. First, is the conformational change caused by the C23–C25 and/or C14–C15 bond cis–trans photoisomerization or other types of distortion? Second, does the ESIPT occur at both HBI halves of the BBM molecule? To address the first question, we obtained time-resolved upconversion fluorescence spectra for BBM following 350 nm excitation (see Figure 3 and Appendix A). As shown in Figure 3a, in DMSO, a blue emission band maximizes at approximately 438 nm at time zero; as time elapsed, the emission band redshifts to a final position nearly corresponding to the steady-state emission spectrum with a maximum of 445 nm. The DADS spectra yield three components including τ_1_ = 0.3 ± 0.2 ps rising dynamics followed by τ_2_ = 2.9 ± 0.3 ps and τ_3_ = 1.3 ± 0.1 ns (see Figure 3b). The ultrafast rise with a time of 0.3 ps reflects the formation of the BBM-keto* component via the ESIPT. In particular, the transient intensity of emission on the blue side of the emission band (420 nm) exhibits an ultrafast rise (~0.3 ps) followed by a dominant fast decay component of 2.9 ps and a long decay lifetime of 1.3 ns (see Figure 3c); at the longest emission wavelength of 480 nm, the kinetics is well-fitted by a fast rise of 2.9 ps and a 1.3 ns single exponential decay component, in which the rise of 2.9 ps matches the decay component of the short emission wavelength at 420 nm and is consistent with the τ_2_ component obtained from the TA spectra. In the THF solvent, as shown in Appendix A, similar excited-state features were observed as DMSO; three components of τ_1_ = 0.3 ± 0.2 ps, τ_2_ = 1.6 ± 0.4 ps, and τ_3_ = 1.2 ± 0.2 ns were required for the best fit.

The transient amplitude and frequency change of the second component τ_2_ in two solvents shows distinct dynamic differences (see Figure 3c, Appendix A). A previous report indicated that the solvent relaxation times of DMSO and THF are 1.8 ps and 0.9 ps, respectively [31]. In our experiment, as depicted in Appendix A, the dynamic Stokes shift of the transient fluorescence frequency in DMSO (3.9 ps) and THF (1.5 ps) exhibits a longer lifetime than their respective solvent relaxation time. This indicates that solvation and conformational change may simultaneously occur until complete solvation is achieved, after which conformational change dynamics primarily govern the excited-state relaxation [21,32]. Additionally, the DADS spectrum of τ_2_ displays a dispersive lineshape indicating a dynamic redshift with enhanced fluorescence intensity (see Figure 3d). This phenomenon cannot be attributed to cis–trans photoisomerization along C23–C25 or C14–C15 bonds in the BBM-keto* form; as such, photoisomerization would effectively quench fluorescence intensity and result in non-radiative decay [6,9]. There is, however, a flexible point at the C4–C7 central bridge that may facilitate conformational changes by rotating the dihedral angle between the two HBI halves of the BBM-keto* form [33,34,35,36].

### 2.4. Femtosecond Raman Stimulated Spectroscopy

To summarize our findings thus far, the TA experimental and upconversion fluorescence results provide evidence for ESIPT and conformational change evolution in the BBM molecule. However, further support is required to precisely assign the τ_1_ and τ_2_ time constants. For τ_1_, does the ESIPT occur at two HBI halves or only one of the HBI half in BBM? For τ_2_, does the conformational change derive from the rotation through the C_4_–C_7_ bond? To gain more experimental insights into conformational change during ESIPT, FSRS was used to track local vibrational marker bands on the excited states [20,21,22,24,25,37,38]. We strategically selected 600 and 480 nm Raman pumps, which correspond to the red shoulder of BBM-keto* and BBM-enol* ESA bands. This pre-resonantly enhances the FSRS signal on the Stokes side (see Appendix A), providing the excited-state Raman signals that reveal intricate dynamics tracking ESIPT and the structural change of BBM molecules [20,21,22]. Figure 4a reveals a 2D contour plot of FSRS spectra for BBM across the frequency range of 850 cm^−1^–1700 cm^−1^ in DMSO with Raman pump excitation of 600 and 480 nm (baseline drawn and FSRS raw data are shown in Appendix A). Five excited-state Raman peaks were observed at 953 cm^−1^ (I), 1130 cm^−1^ (II), 1358 cm^−1^ (III), 1565 cm^−1^ (IV), and 1587 cm^−1^ (V) under the Raman pump excitation of 480 nm (BBM-enol* form). A more detailed description of Raman modes at the BBM-enol form was achieved through density functional theory (DFT) with the optimized geometry at the dihedral angles between the two HBI halves at Θ = 39° and between benzene and imidazole rings at Φ = 0°. As illustrated in Appendix A and Appendix A, the observed Raman modes at 1565 cm^−1^ and 1587 cm^−1^ are mainly attributed to the C14–C15 (C23–C25) stretching accompanied by the benzene and imidazole ring deformation; the mode at 1358 cm^−1^ is assigned to the benzene and imidazole ring stretching and C–H in-plane waging; 1130 cm^−1^ is attributed to C–H in-plane waging motions of two benzene rings; 953 cm^−1^ is attributed to C–H out-of-plane bending of two benzene rings. Those modes shift to 939 cm^−1^ (I), 1122 cm^−1^ (II), 1348 cm^−1^ (III), 1534 cm^−1^ (IV), and 1568 cm^−1^ (V) at the Raman pump of 600 nm (BBM-keto* form).

Figure 4b presents the transient amplitude of Raman modes (IV) and (V) of BBM in DMSO at different Raman pump wavelengths. Upon the excitation with a 480 nm Raman pump, both Raman modes exhibit two exponential decays with a time constant of τ_1_ = 320 ± 150 fs and an infinitely long lifetime. τ_1_ is consistent with the ESIPT rate observed in TA and upconversion fluorescence, indicating that it resonantly enhances the excited-state Raman signal of the BBM-enol* form under the excitation of a 480 nm Raman pump; the subsequent infinitely long lifetime can be attributed to the decay of fluorescence emission from the BBM-enol* state. By using a 600 nm Raman pump, the transient amplitude of these two Raman modes exhibits a rising dynamics of τ_1_ = 260 ± 90 fs followed by two exponential decays with time constants of τ_2_ = 3.6 ± 0.5 ps and an infinitely long lifetime. Moreover, Appendix A illustrates that τ_2_ displays significant viscosity-dependent dynamics in THF (2.6 ± 0.4 ps) and corroborates the aforementioned conformational change dynamics observed in TA and fs upconversion spectra. Importantly, the Raman mode (IV) and mode (V) can be mainly attributed to two similar stretching modes, namely C14–C15 and C23–C25 stretching modes, which are separately located in two HBI moieties of the BBM molecule, as shown in Figure 4c; the difference in the peak frequency between modes (IV) and (V) is 22 cm^−1^ in the BBM-enol* form (with a Raman pump of 480 nm), which increases to 34 cm^−1^ in the BBM-keto* form (with a Raman pump of 600 nm). As presented in Appendix A and Appendix A, the DFT calculations demonstrate that the peak frequency difference of the two Raman modes in the BBM-enol form is 4 cm^−1^; a slight decrease was observed in the BBM-keto form (3 cm^−1^) when the ESIPT occurs in both HBI halves. Conversely, if the ESIPT only takes place in one HBI half of BBM, there is a significant increase in the peak frequency difference from 4 cm^−1^ in the BBM-enol* form to 38 cm^−1^ in the BBM-keto* form (see Appendix A and Appendix A). The observed trend from the calculation is consistent with experimental findings, indicating that the symmetrical structure of the BBM-enol form containing two HBI-enol moieties is disrupted in the BBM-keto form (containing one HBI-enol and one HBI-keto form), resulting in greater frequency separation between these two stretching modes. Additionally, Figure 4c demonstrates that the FSRS spectra at 480 nm Raman pump excitation exhibit equal peak intensity for the two Raman modes (IV) and (V) with a Gaussian line shape in the BBM-enol* form. However, in the BBM-keto* form (600 nm Raman pump), there is a difference in peak intensity between these two Raman modes due to ESIPT occurring in one HBI half of BBM, resulting in different Raman polarizability.

Figure 4d illustrates the transient frequency shift of modes (I) and (II) in the BBM-keto* form with Raman pump excitation of 600 nm. The 934 cm^−1^ mode rapidly blueshifts to 941 cm^−1^ in 0.3 ps, followed by a slow redshift to 936 cm^−1^ with a time constant of 3.8 ps, while mode (II) exhibits two exponential frequency blueshifts from 1117 cm^−1^ to 1128 cm^−1^ with time constants of 0.3 ps and 3.8 ps. The initial blueshift of the two Raman modes can be attributed to the vibrational cooling process accompanied by ESIPT. To investigate the second transient frequency change in those modes, TD-DFT calculations were performed for the BBM-keto* form with ESIPT occurring in one HBI half of BBM. The optimized geometry at the S1 electronic excited state of BBM-keto* reveals the dihedral angle of Θ = 26° and Φ = ~0°. By reducing Θ from 39° (the dihedral angle on the S1 Franck–Condon state) to 26° while keeping Φ = 0°, it was observed that the Raman modes (I) and (II) exhibit redshift and blueshift, respectively, during the twisting of Θ (see Appendix A). This suggests that the second transient frequency change is caused by the rotation of the dihedral angle Θ between the two halves of BBM, ultimately leading to a planarized BBM-keto* isomer.

At last, to clarify the electron transitions responsible for the dynamic redshift fluorescence spectra observed in fs fluorescence upconversion spectra, we used TD-DFT to calculate the energy gap and oscillator strength of the S1 → S0 transition of the BBM-keto* form (ESIPT occurs in one HBI half of BBM) at different torsion angles Θ = 26°, 30°, and 35° in the S1 excited state in the DMSO solvent. The results presented in Appendix A demonstrate an evident fluorescence spectral redshift and an increase in oscillator strength as Θ is tuned from 39° to 26°. This finding is consistent with the experimental observation of fs fluorescence upconversion spectra, further confirming that the planarization via the Θ dihedral angle torsion plays a crucial role in the excited-state relaxation of the BBM keto* state.

## 3. Materials and Methods

### 3.1. Synthesis of BBM

As shown in Appendix A, NaHSO_3_ (0.97 g, 9.33 mmol) and 2,4-dimethylbenzaldehyde (1.29 g, 9.33 mmol) were dissolved in 50 mL ethanol. Moreover, the resultant mixture was stirred at room temperature for 12 h. A DMF solution (50 mL) containing 3,3-diaminobenzidine (1 g, 4.67 mmol) was then added, and the mixture was continuously stirred under reflux for 12 h at 80 °C. After cooling to room temperature, vacuum distillation was available to remove most of the solvents, and the precipitate was obtained by adding 100 mL of deionized water. The compound was collected by filtration and washed with deionized water. The resulting solid was dried for 3 days to afford the production with a yield of 71%.

### 3.2. Transient Absorption Spectroscopy

Transient absorption (TA) spectroscopy was carried out on commercial transient absorption (HELIOS, Ultrafast System LLC, Sarasota, FL, USA). The 800 nm laser pulse was generated from a commercial mode-locked Ti: sapphire laser (Coherent Inc., Santa Clara, CA, USA, Astralla tunable USP, 35 fs, 7 mJ/pulse, 1 kHz repetition rate). A 350 nm pulse was used as the actinic pump to stimulate the sample at TA experiment, which was generated by 800 nm laser passing an optical parametric amplifier (OPerA solo, Coherent Inc., USA); the pulse duration of the output from the OPA is about 60 fs. A white light continuum ranging from ~350 nm to ~650 nm serves as the probe pulse, which was obtained by focusing on a 2 mm thick sapphire. The instrument response function (IRF) of this TA system was determined to be 120 fs.

EOS-TA spectroscopy was carried out on commercial transient absorption (HELIOS, Ultrafast System). A 350 nm pulse generated by 800 nm light source passing an optical parametric amplifier (OPerA solo, Coherent Inc.) was used as the actinic pump. A broadband supercontinuum white light with a wavelength range from ~350 nm to ~800 nm serves as the probe pulse, which was generated through a sub ns white light laser.

### 3.3. Time-Resolved Upconversion Fluorescence Spectroscopy

The fs fluorescence upconversion spectroscopic measurements were performed using a commercial time-resolved fluorescence upconversion spectrometer (Halcyone Fire, Ultrafast Systems LLC, USA). The excited wavelength of fluorescence upconversion experiments was selected at 350 nm as an actinic pump. The gate pulse comes from a small portion of an 800 nm fundamental beam. The fluorescence generated by the sample is collected and interacts with the gate pulse to produce a sum frequency signal on the BBO crystal. Upconversion signals were gathered using a monochromator and detected using a CCD camera. The time resolution was determined to be ~250 fs.

### 3.4. Femtosecond Stimulated Raman Spectroscopy (FSRS)

The fundamental laser pulses (Astrella, Coherent Inc., 35 fs pulse duration, 7 mJ pulse energy, and 1 kHz repetition rate) were split into three beams to generate a tunable narrowband picosecond (ps) Raman pump, a broadband femtosecond (fs) Raman probe, and a fs actinic pump. The actinic pump beams centered at 350 were generated by an optical parametric amplifier/OPA (OPerA Solo, Coherent Inc.) for excited-state FSRS. About 3 W of fundamental pulses was directed through a second harmonic bandwidth compressor (SHBC, Coherent, Inc.) and a ps-OPA system (TOPAS-400, Coherent, Inc.) to generate ps pulses as the Raman pump (300 nJ pulse energy). About 15 mW of the fundamental laser output was focused onto a 2 mm thick single-crystal sapphire plate to obtain the supercontinuum white light as the Raman probe. The instrument response time measured by the cross-correlation between the fs actinic pump and Raman probe pulses was ~150 fs.

### 3.5. Calculations

Density functional theory (DFT) and time-dependent DFT (TD-DFT) computations were carried out at the level of PBE1PBE/6-311 + G** by using Gaussian 09 package (cite: Gaussian 09, Revision E.01, M. J. Frisch, et al. Gaussian, Inc., Wallingford, CT, USA, 2013.) The ground state and excited states were geometrically optimized. Vibrational frequencies were calculated on the optimized geometries. The solvation effect was described by the polarizable continuum model (PCM) [39,40,41,42,43,44].The atomic coordinates is exhibited in Appendix A.

## 4. Conclusions

In this study, we exploit several ultrafast spectroscopic techniques across the electronic and vibrational domains to demonstrate the light-induced ultrafast ESIPT and structural dynamics of the BBM molecules in different solvents. Building upon all the experimental and calculation results, we can draw a schematic diagram of the potential energy surface summarizing the photo-induced structural dynamics observed in BBM (see Figure 5). Upon the vertical excitation of BBM-enol to its excited-state BBM-enol*, it undergoes ESIPT in one HBI half of BBM, resulting in the formation of the deprotonated BBM-keto* state (300 fs time constant). This is followed by the decay of BBM-keto* to ground-state BBM-keto via 438 nm fluorescence (with a lifetime of 1.3 ns) and simultaneously planarization through variation in the dihedral angle Θ between the two HBI halves, resulting in the dynamic fluorescence redshift to 445 nm. This work demonstrates that FSRS, after successful assignment by quantum chemical calculation, especially when combined with the time-resolved electronic spectroscopies, is a powerful method for uncovering the ultrafast and small structural changes of the chromophore under the effect of different solvent environments.

## Figures and Tables

**Figure 1 ijms-24-09438-f001:**
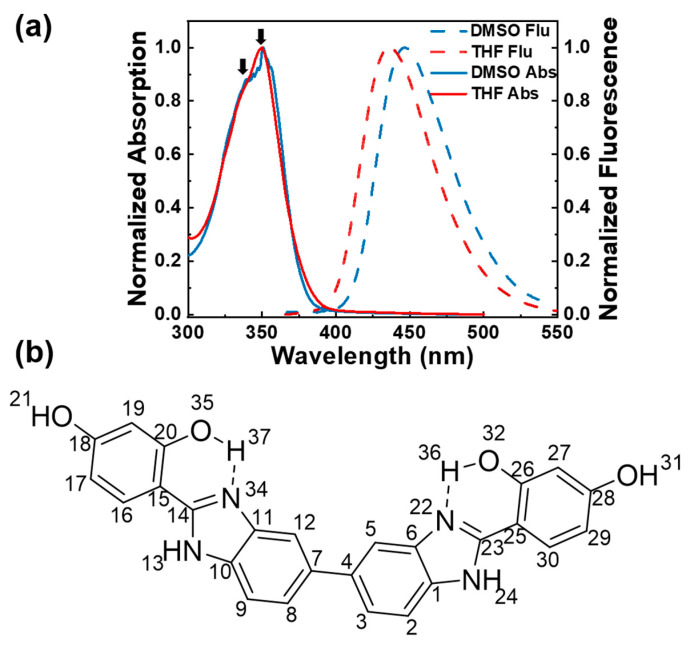
(**a**) Steady-state absorption and fluorescence spectra of BBM in DMSO and THF; (**b**) molecular structure of BBM-enol form, arrows point the absorption peaks.

**Figure 2 ijms-24-09438-f002:**
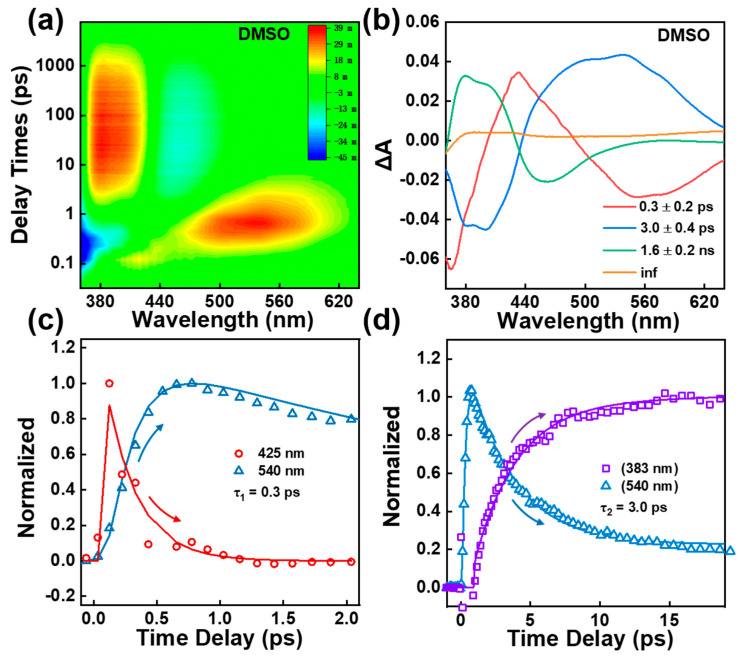
(**a**) 2D plot of TA spectrum of BBM in DMSO at 350 nm excitation (time delay is plotted using log scale). (**b**) Decay-associated difference spectra (DADS) of TA spectrum of BBM in DMSO. (**c**) Transient amplitude of TA spectra of BBM probe at 425 nm and 540 nm in DMSO; τ1 is retrieved from global analysis. (**d**) Transient amplitude of TA spectra of BBM probe at 383 nm and 540 nm in DMSO; τ_2_ is retrieved from global analysis.

**Figure 3 ijms-24-09438-f003:**
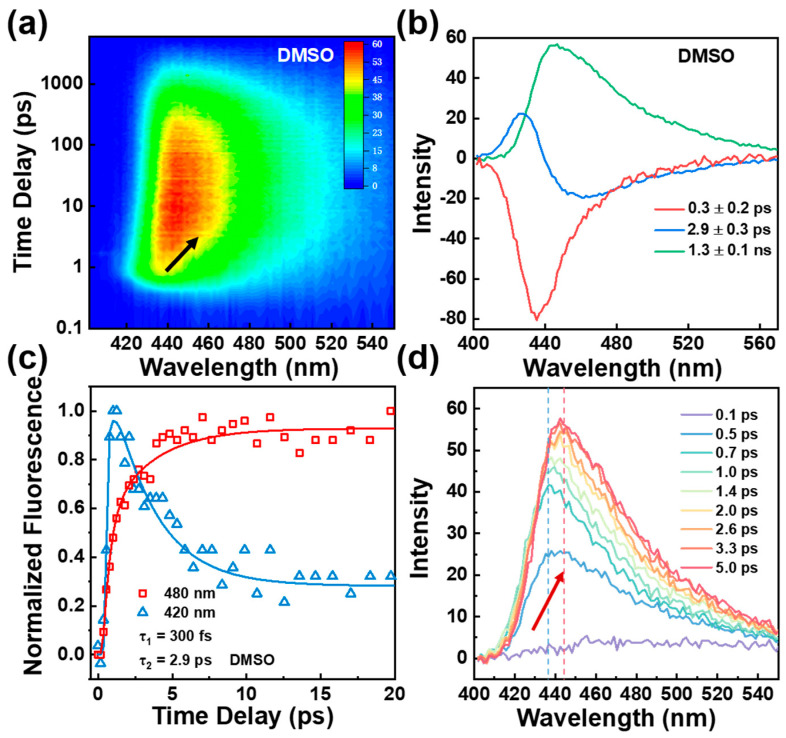
(**a**) 2D plot of fs fluorescence upconversion spectra of BBM under 350 nm excitation in DMSO (time delay is plotted using log scale). (**b**) DADS spectrum of fs fluorescence upconversion of BBM in DMSO. (**c**) Transient amplitude of BBM fluorescence probe at 420 nm and 480 nm in DMSO; τ_1_ and τ_2_ are retrieved from global analysis. (**d**) fs fluorescence upconversion spectra of BBM in DMSO at different delay times.

**Figure 4 ijms-24-09438-f004:**
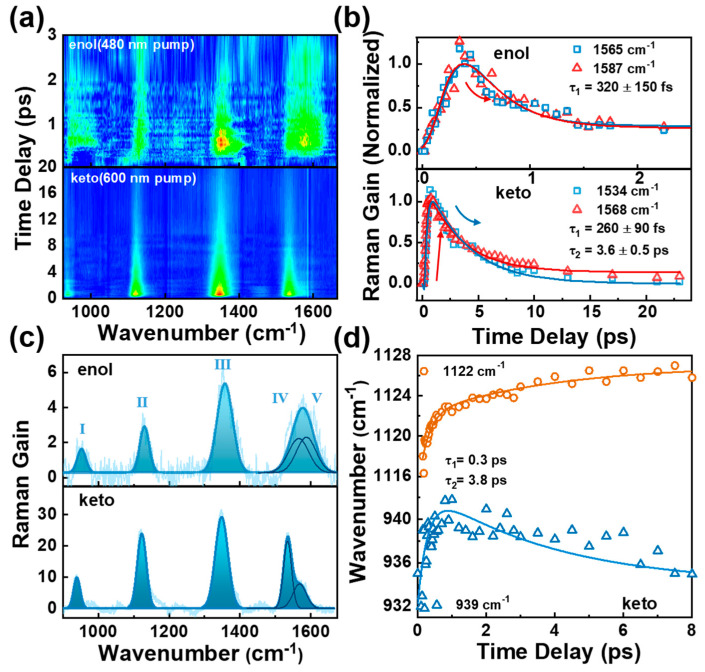
(**a**) FSRS 2D plot of BBM stimulated by 350 nm, resonantly enhanced by 480 nm and 600 nm in DMSO. (**b**) Transient amplitude of Raman modes (IV) and (V) with 480 nm and 600 nm Raman pump. (**c**) FSRS spectra of BBM-enol* and BBM-keto* forms. (**d**) Transient frequency of the Raman modes (I) and (II) of BBM-keto* in DMSO.

**Figure 5 ijms-24-09438-f005:**
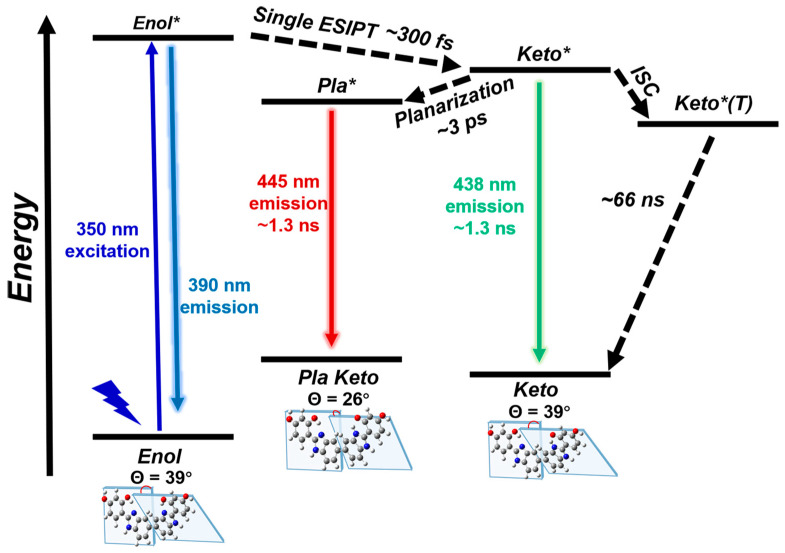
Schematic representation of photodynamic of BBM in DMSO. Pla (planarized), ESIPT (excited-state intramolecular proton transfer), ISC (intersystem crossing), * (excited state).

## Data Availability

Not applicable.

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
