# Peer review of "Investigation of Excited-State Intramolecular Proton Transfer and Structural Dynamics in Bis-Benzimidazole Derivative (BBM)"

_ijms, 2023, doi:10.3390/ijms24119438_

Round 1

Reviewer 1 Report

Review Report:

In the manuscript, Xie et al., report the enol to keto excited state intramolecular proton transfer (ESIPT) in bis-benzimidazole derivative (BBM) by employing transient absorption, fs fluorescence up-conversion, fs stimulated Raman spectroscopy (FSRS) and TDDFT calculations. Their finding clearly demonstrates that ESIPT occurs in an ultrafast timescale (300 fs), and it is only limited to a single hydroxyphenyl benzimidazole subunit. Through FSRS they further reveal that the structural change occurs through the pivotal bond connecting these two subunits.

I find the work important in the context of light induced photochemistry. The results are consistent and thorough Therefore, I recommend publication of the present work. However, I find a few minor issues in the manuscript, which need to be addressed. I have used the following abbreviations, P-page number and L-line number to point out my comments.

Specific Comments:

1. P2-L67: Hydroxypheny should be written as hydroxyphenyl.

2. P2-Figure 1: I find the absorption spectrum of BBM in DMSO and THF are different. In THF, I do not see any shoulder as in DMSO. I encourage the authors to provide an explanation for this difference.

In Figure 1, I noticed the major and minor ticks are directed outwards in the y-axis, whereas in rest of the figures those are directed inward. I recommend the authors to be consistent with the figures.

3. P3-L81: I recommend the authors to use pulse energy instead of power intensity.

4. P3-Figure 2(a): The y-axis is not linear right. The authors should point that out in the Figure caption. I find this similar thing in Figure 3(a) and some figures in supporting material. I encourage the authors to fix this.

5. P3-Figure 2(b): I recommend the authors to use same line color for the comparable lifetimes in DMSO and THF. Then it will be easier for comparison.

The same is true for Figure 3(b).

6. P4-L109: The authors should mention that BBM-keto is in the excited state.

8. P6-L210: I find a significant discrepancy in Raman modes between the experimental and theoretical values in BBM-enol* with 480 nm pump. It is experimentally 22 cm-1 while theoretically it is 4 cm-1. While this difference closely agrees in the 600 nm pump. An explanation for this is expected.

9. P8-L298: Did the authors use any scaling factor while reporting the vibrational frequencies?

General Comments:

1. I encourage the authors to report xyz coordinates of the geometries and the corresponding vibrational frequencies in addition to the oscillator strength for the electronic transitions in various solvents.

2. The authors are advised to report the error bars while reporting the life times.

Author Response

Response to Reviewer#1

We sincerely appreciate reviewer 1’s comment “I find the work important in the context of light-induced photochemistry. The results are consistent and thorough”. Thanks for your time and efforts expended in an attempt to improve our paper. We have revised our manuscript in accordance with your valuable suggestions.

Comment 1: P2-L67: Hydroxypheny should be written as hydroxyphenyl.

Reply: Thank you very much for your careful reading. We have corrected the spelling of hydroxyphenyl.

Comment 2. P2-Figure 1: I find the absorption spectrum of BBM in DMSO and THF are different. In THF, I do not see any shoulder as in DMSO. I encourage the authors to provide an explanation for this difference.

Reply: Thank you very much for your valuable questions. Indeed, the peak shoulder of anti-enol in THF is not as clear as in DMSO. This phenomenon can be explained as the difference in a solvent-dependent conformational equilibrium between the syn- and anti-enols, which induced the different proportions of syn- and anti-enols.

To clarify this point, the sentence “Notably, the peaks corresponding to the anti-enol in THF exhibits less distinctness than that observed in DMSO. This phenomenon can be attributed to a solvent-dependent conformational equilibrium between syn- and anti-enols, which results in varying proportions of these species.” As added in L68-71.

Comment 3: In Figure 1, I noticed the major and minor ticks are directed outwards in the y-axis, whereas in rest of the figures those are directed inward. I recommend the authors to be consistent with the figures.

Reply: Thank you very much for your valuable advice. We have changed the y-axis of Fig.1 to be consistent with other figures.

Comment 4: P3-L81: I recommend the authors to use pulse energy instead of power intensity.

Reply: Thank you very much for your valuable advice. We have changed the “power intensity” to” pulse energy”.

Comment 5. P3-Figure 2(a): The y-axis is not linear right. The authors should point that out in the Figure caption. I find this similar thing in Figure 3(a) and some figures in supporting material. I encourage the authors to fix this.

Reply: Thank you very much for your valuable advice. We have added the sentence “y-axis is plotted using log scale” in the Figure caption of Figure 2(a), Figure 3(a), and the figures in supporting material.

Comment 6. P3-Figure 2(b): I recommend the authors to use same line color for the comparable lifetimes in DMSO and THF. Then it will be easier for comparison.The same is true for Figure 3(b).

Reply: Thank you very much for your valuable advice. We have changed Figure S2 and Figure S6(b) to ensure the color for the comparable lifetimes in DMSO and THF is the same.

Comment 7. P4-L109: The authors should mention that BBM-keto is in the excited state.

Reply: Thank you very much for your careful reading. We have changed the sentence “ BBM-keto triplet state T1 state” to “BBM-keto* triplet state T1 state” in L114.

Comment 8. P6-L210: I find a significant discrepancy in Raman modes between the experimental and theoretical values in BBM-enol* with 480 nm pump. It is experimentally 22 cm-1 while theoretically it is 4 cm-1. While this difference closely agrees in the 600 nm pump. An explanation for this is expected.

Reply: Thank you for your question. According to the DFT calculation, the Raman modes observed at 1565 cm-1 and 1587 cm-1 can be mainly attributed to two similar stretching modes, namely C14-C15 and C23-C25 stretching modes, which are separately located in two HBI moieties of BBM molecule. Our DFT results indicate a slight difference in peak frequency between these two modes, which can be attributed to the symmetric structure of the BBM-enol form containing two HBI-enol moieties. While for the BBM-keto form which contains one HBI-enol and HBI-keto forms, the “symmetry” of BBM is broken, in which these two stretching modes become more “non-degenerate”. So the trend is consistent with that from experimental observation.

The discrepancy between the experimentally observed and calculated Raman peak difference of modes VI and V can be attributed to differences in molecular geometry resulting from the use of optimized geometries in DFT calculations versus actual experimental conditions.

Comment 9. P8-L298: Did the authors use any scaling factor while reporting the vibrational frequencies?

Reply: Thank you very much for your valuable comment. The calculation results presented in this manuscript are reported using the original numbers without any scaling factors applied.

General Comments:

  1. I encourage the authors to report xyz coordinates of the geometries and the corresponding vibrational frequencies in addition to the oscillator strength for the electronic transitions in various solvents.

Reply: Thank you for your valuable advice. We have added the optimized structure of BBM-enol, BBM-keto-pt2, and BBM-keto-pt1 Raman modes (I-V) and corresponding vibrational frequencies in Figure S11-S13 and Table S1. Furthermore, we have provided the atomic coordinates of DFT calculation BBM in Table S2.

In this manuscript, we have employed TD-DFT to calculate the oscillator strength of electronic transitions in DMSO solvents. The primary objective of the calculation is to further confirm that the planarization via Θ dihedral angle torsion plays a crucial role in the excited state relaxation of BBM keto* state. As stated in the manuscript, while solvents may affect energy gap and oscillator strength values, the primary factor contributing to the observed fluorescence dynamics red-shift from fluorescence up-conversion is the conformational change (planarization). Therefore, we only performed TD-DFT calculation in DMSO solvent and obtained results consistent with experimental observations. To clarify this point, we have added "in DMSO solvent" in L247.

  1. The authors are advised to report the error bars while reporting the life times.

Reply: Thank you for your valuable advice. We have incorporated error bars into the DADS global fitting results for both transient absorption and time-resolved fluorescence measurements. Additionally, we utilized the DADS global fitting results to determine the time constants of kinetic traces at single probe wavelengths (e.g., Fig 2c,d and Fig 3c). Furthermore, we have included error bars in our FSRS data analysis.

Reviewer 2 Report

The present work concerns the investigation of Excited State Intramolecular Proton Transfer and structural dynamics in Bis-benzimidazole Derivative molecule (BBM) using some time-resolved spectroscopy techniques. (Transient absorption, fluorescence upconversion, and femtosecond stimulated Raman spectroscopy). As such, it is a stimulating topic; however, the work presents several criticisms and needs to be improved to increase its quality. 

First, the title is not very appropriate, it should be more general because in this study the authors use several ultrafast techniques, not only femtosecond stimulated Raman spectroscopy but also time-resolved electronic spectroscopies, aided by quantum chemical calculations to investigate the photodynamics of BBM molecule.

From my point of view, one of the main problems of the manuscript is that it lacks a proper discussion of the obtained results. The manuscript conclusions are not properly corroborated by the experimental and computational results.

Additional work regarding the results and especially the discussion section is necessary to make the manuscript acceptable for publication. 

Here are my main comments on the manuscript:

1.     The introduction can be improved by clarifying and rephrasing.

lines 29-31: “As molecules binding to the metal cations are highly sufficient …. emission”

line 46 “Raman loss femtosecond stimulated Raman technique.”

lines 55-60 this sentence is the conclusion of the manuscript.

      2.   The result and discussion sections should be significantly revised:

 a) the authors (lines 66-68) write about anti-enol and syn-enol (not cis) BBM conformers, but this equilibrium is not considered in the discussion of the time-resolved spectroscopies. Why?

b) line 73: the effect of solvent polarity on the excited electron states.

Several studies have documented the solvent-dependence conformational isomerism of derivatives of 2-(2’-hydroxyphenyl) benzimidazole (HBI) and its analogs 2-(2’- hydroxyphenyl) benzoxazole (HBO) and 2-(2’-hydroxyphenyl) benzothiazole (HBT) and the solvent dependent dynamics (enol dynamics, proton transfer dynamics, and keto dynamics)

Why did the authors not characterize the effect of solvation on the steady-state spectroscopic properties of BBM in different solvents, such as hexane and MeOH?

         c) there are several errors in the values of the decay times in the text  and figures. Check carefully.

 d) the authors must clearly and accurately discuss the results obtained from different techniques.

Check fig.2d

            Lines 150- 158 improve the clarity

Lines 213-217 why?

Author Response

Response to Reviewer#2

Thanks for your time and efforts expended in the attempt to improve our paper, and we appreciate your comment “it is a stimulating topic”. We are glad that you are interested in our work. We have revised our manuscript in accordance with your valuable suggestions.

At first, we changed the title to be more general, “Investigation of Excited State Intramolecular Proton Transfer and Structural Dynamics in Bis-benzimidazole Derivative (BBM) ”.

Comment 1.The introduction can be improved by clarifying and rephrasing.

lines 29-31: “As molecules binding to the metal cations are highly sufficient …. emission”

line 46 “Raman loss femtosecond stimulated Raman technique.”

lines 55-60 this sentence is the conclusion of the manuscript.

Reply: Thank you very much for your useful advice.

  1. the sentence in lines 29-31 was changed to “When the molecules bind sufficiently to the metal cations, these cations act as ligands and can disrupt the ESIPT process, resulting in a decrease in keto emission and an increase in enol fluorescence emission simultaneously”

2.We have changed the “ Raman loss femtosecond stimulated Raman technique” to “ultrafast Raman loss spectroscopy (URLS)”.

3 the sentence in lines 55-60 was truncated “Different from the HBI molecule, BBM exhibits unique ESIPT and intramolecular twisting dynamics.”.

Comment 2. The result and discussion sections should be significantly revised:

  1. a) the authors (lines 66-68) write about anti-enol and syn-enol (not cis) BBM conformers, but this equilibrium is not considered in the discussion of the time-resolved spectroscopies. Why?

Reply: Thank you very much for your careful reading and valuable question, and we corrected “cis-enol” to “syn-enol”. The absorption peak of anti-enol is situated at the blue edge of the syn-enol conformer. Therefore, selecting 350 nm as an actinic pump to excite the BBM results in low excitation efficiency for anti-enol conformer. Moreover, since BBM predominantly favors syn-enol conformation, this further reduces the time-resolved spectroscopic signals of anti-enol in its excited state. Secondly, the ESIPT is absent the in anti-enol conformer, hence we did not discuss the anti-enol in this manuscript.

  1. b) line 73: the effect of solvent polarity on the excited electron states.

Several studies have documented the solvent-dependence conformational isomerism of derivatives of 2-(2’-hydroxyphenyl) benzimidazole (HBI) and its analogs 2-(2’- hydroxyphenyl) benzoxazole (HBO) and 2-(2’-hydroxyphenyl) benzothiazole (HBT) and the solvent dependent dynamics (enol dynamics, proton transfer dynamics, and keto dynamics)

Why did the authors not characterize the effect of solvation on the steady-state spectroscopic properties of BBM in different solvents, such as hexane and MeOH?

Reply: Thank you very much for your valuable question. We have attempted to dissolve BBM using various solvents. However, the solubility of the BBM sample in most of the solvents (including hexane) is insufficient for conducting TA and FSRS experiments. While MeOH can effectively dissolve BBM, its protic nature causes a blueshift in the steady-absorption peak compared to that of BBM in DMSO and THF solvents, which is attributed to the formation of a strong hydrogen bond between BBM and MeOH. Furthermore, we have explored transient absorption for BBM in MeOH, the results showcase distinct dynamics compared to that observed from BBM in DMSO and THF solvents. only a decay dynamics of BBM- Keto* was obtained in BBM - MeOH.

  1. c) there are several errors in the values of the decay times in the text and figures. Check carefully.

Reply: Thank you very much for your careful reading. We have corrected the values.

  1. d) the authors must clearly and accurately discuss the results obtained from different techniques.

Check fig.2d

Lines 150- 158 improve the clarity

Lines 213-217 why?

Reply: 1. Thank you very much for your careful reading, we have checked the Fig. 2d carefully. The lifetime in Fig.2d of 3.1 ps is corrected to 3.0 ps.

  1. To clarify the point, the sentence in Lines 150-158 was changed to “The previous report indicated that the solvent relaxation time of DMSO and THF are 1.8 ps and 0.9 ps, respectively. In our experiment, as depicted in Fig. S8, the dynamic stokes shift of the transient fluorescence frequency in DMSO (3.9 ps) and THF (1.5 ps) exhibits a longer lifetime than their respective solvent relaxation time. This indicates that solvation and conformational change may occur simultaneously until complete solvation is achieved, after which conformational change dynamics primarily govern the excited state relaxation”
  2. To clarify the point, sentences in Line150-158 were revised to “Importantly, the Raman mode (IV) and mode (V) can be mainly attributed to two similar stretching modes, namely C14-C15 and C23-C25 stretching modes, which are separately located in two HBI moieties of BBM molecule, as shown in Fig. 4(c), the difference in peak frequency between modes (IV) and (V) is 22 cm-1 in BBM-enol* form (with Raman pump of 480 nm), which increases to 34 cm-1 in BBM-keto* form (with Raman pump of 600 nm). As presented in Table. S1, the DFT calculations demonstrate the peak frequency difference of the two Raman modes in BBM-enol form is 4 cm-1, a slight decrease was observed in BBM-keto form (3 cm-1) when the ESIPT occurs in both HBI halves. Conversely, if the ESIPT only takes place in one HBI half of BBM, there is a significant increase in the peak frequency difference from 4 cm-1 in BBM-enol* form to 38 cm-1 in BBM-keto* form. The observed trend from the calculation is consistent with experimental findings, indicating that the symmetrical structure of the BBM-enol form containing two HBI-enol moieties is disrupted in the BBM-keto forms (containing one HBI-enol and one HBI-keto form), resulting in greater frequency separation between these two stretching modes.”

Round 2

Reviewer 2 Report

The paper was improved and the authors addressed adequately all my comments.